

# BioInstaller: a comprehensive R package to construct interactive and reproducible biological data analysis applications based on the R platform

Jianfeng Li[1,*], Bowen Cui[1,*], Yuting Dai[1,2], Ling Bai[1] and Jinyan Huang[1]

[1] State Key Laboratory of Medical Genomics, Shanghai Institute of Hematology, National Research Center for Translational Medicine, Ruijin Hospital, Shanghai Jiao Tong University School of Medicine, Shanghai Jiao Tong University, Shanghai, China
[2] School of Life Sciences and Biotechnology, Shanghai Jiao Tong University, Shanghai, China
* These authors contributed equally to this work.

## ABSTRACT

The increase in bioinformatics resources such as tools/scripts and databases poses a great challenge for users seeking to construct interactive and reproducible biological data analysis applications. Here, we propose an open-source, comprehensive, flexible R package named BioInstaller that consists of the R functions, Shiny application, the HTTP representational state transfer application programming interfaces, and a docker image. BioInstaller can be used to collect, manage and share various types of bioinformatics resources and perform interactive and reproducible data analyses based on the extendible Shiny application with Tom's Obvious, Minimal Language and SQLite format databases. The source code of BioInstaller is freely available at our lab website, http://bioinfo.rjh.com.cn/labs/jhuang/tools/bioinstaller, the popular package host GitHub, https://github.com/JhuangLab/BioInstaller, and the Comprehensive R Archive Network, https://CRAN.R-project.org/package=BioInstaller. In addition, a docker image can be downloaded from DockerHub (https://hub.docker.com/r/bioinstaller/bioinstaller).

## INTRODUCTION

With the rapid development of new bioscience technology, particularly next-generation sequencing (NGS), volumes of "omics" data have been generated, such as the 1000 Genomes Project, The Cancer Genome Atlas (TCGA), and Genotype-Tissue Expression (*Abecasis et al., 2012*; *Cancer Genome Atlas Research Network et al., 2013*; *Lonsdale et al., 2013*; *Sanchez-Vega et al., 2018*). The bioinformatics tools and databases required for the downstream data analysis are also increasing at a phenomenal rate. R language, as the most popular programming language for statistics, biological data analysis, and big data, has enabled diverse and free R packages (>14,000) for different types of applications, such as high-throughput sequencing data analysis (e.g., Bioconductor)

Corresponding author
Jinyan Huang, jinyan@shsmu.edu.cn

(*Gentleman et al., 2004*) and the development of web applications (e.g., Shiny framework) (*Chang et al., 2015*). With the development of web technologies and the release of the R web developmental framework Shiny, the number of interfaces available to R users has increased. However, due to the lack of high-performance and open-source cloud platforms based on R (e.g., Galaxy for Python users) (*Afgan et al., 2016*), it is still difficult for R users, especially those without web development skills, to construct interactive and reproducible biological data analysis applications supporting the upload and management of files, long-time computation, task submission, tracking of output files, exception handling, logging, export of plots and tables, and extendible plugin systems.

Another common problem usually faced by R and other programming platform users (e.g., the team of Galaxy) (*Afgan et al., 2016*) is how to acquire and share certain bioinformatics resources quickly and accurately. Numerous bioinformatics tools (e.g., primer design, sequence alignment, variant calling and annotation) or scripts (e.g., data format conversion, text processing) are scattered around world web hosts. Biomedical databases are facing the same situation. For example, genome sequences (e.g., hg19/hg38 for human, mm9 and mm10 for mouse) are mainly deposited in the UCSC Genome Browser and National Center for Biotechnology Information (https://www.ncbi.nlm.nih.gov/) (*Tyner et al., 2017*). The best-known gene and transcript annotation resources are provided by GENCODE and the RefSeq database (*Derrien et al., 2012*; *O'Leary et al., 2016*). Genetic variants annotation databases, mainly cancer and Mendelian disorder related, are hosted by the original projects (for example, TCGA) and various down-stream tools (for example, ANNOVAR, Variant Effect Predictor and Oncotator) (*McLaren et al., 2016*; *Ramos et al., 2015*; *Wang, Li & Hakonarson, 2010*). Bioconductor is a popular bioinformatics R community for sharing genetic variants and other types of bioinformatics annotation databases via R package (*Gentleman et al., 2004*), but it is difficult for users to share many types of tools/scripts and databases if they do not have the capability of packing their own tools/scripts and databases. In most cases, these resources are isolated and can only be accessed via a command line tool such as rsync (https://rsync.samba.org/) or wget (http://www.gnu.org/software/wget/), to request the corresponding uniform resource locators (URLs). Software distribution tools that do not demand root privileges, such as conda (https://github.com/conda/conda) and spack (*Gamblin et al., 2015*), have greatly improved the acquisition of bioinformatics software. However, considering the huge growth of tools/scripts and databases required for bioinformatics data analysis, the resources supported by these software distribution tools are far from sufficient. Users also need more experience to use these different package management tools under command line environment.

Here, we present an open-source, comprehensive, flexible bioinformatics platform named BioInstaller that can be used to collect, manage and share various types of bioinformatics resources and to perform interactive and reproducible data analyses. By utilizing a simplified and standard Tom's Obvious, Minimal Language (TOML) format configuration file with extra parse functions, the developers and users can freely and unreservedly share their public or internal bioinformatics tools/scripts and databases online on the GitHub repository or other hosts. In addition, users can easily obtain

access to pooled bioinformatics resources via the diverse interfaces of BioInstaller, which includes R functions, the Shiny application (*Chang et al., 2015*) and HTTP representational state transfer (REST) application programming interfaces (APIs) that are rarely adopted in other similar tools. As a practical demonstration, we collected 157 tools/scripts and 110 databases specifically related to genetic variants annotation using the BioInstaller-defined configuration files. Notably, we developed a Shiny application to support functions including system monitoring, the logging system, file management, the queue system, and so on. This application can easily be reused in other Shiny applications. We expect the BioInstaller package and the practices in this work to reduce the difficulty of constructing the interactive and reproducible biological data analysis applications for R users, and to further improve the interactivity and reproducibility of bioinformatics data analysis.

## MATERIALS AND METHODS

### Design and development of BioInstaller

BioInstaller was designed as an interactive R package to collect, manage, and share various types of bioinformatics resources and perform interactive and reproducible data analyses. BioInstaller contains the R functions and the Shiny application (*Chang et al., 2015*) and REST APIs (Fig. 1). Both R and other programming platform users can utilize the functions of BioInstaller, such as by downloading bioinformatics tools/scripts and databases and performing statistical analysis and visualization. The R and Shiny interfaces of BioInstaller were mainly developed in R language and utilize the HTML/CSS and JavaScript languages. To run an instance of BioInstaller, the R program and extra dependent R packages are required. Travis CI (https://www.travis-ci.org/) was used to automatically test the R functions on Linux and MAC OSX platforms. Periodically, the tested and updated BioInstaller package is submitted to Comprehensive R Archive Network (CRAN) with an increased version number, for example, from v3.3.3 to v3.3.4. Both the open and restricted bioinformatics resources can be integrated using the TOML format configuration file. The configuration files can also be used in other programming language platforms to access desired masteries by using a unique item name, such as "bwa," "gatk," "annovar," "db_annovar_1000g," "db_annovar_gtex," etc. A hash value was generated using the item name and version for the unique ids of tools/scripts and databases. An autogenerated docker image containing all required R packages and the backend web service of BioInstaller have been deposited at the DockerHub (https://hub.docker.com/r/bioinstaller/bioinstaller).

### GitHub API and custom values/functions for querying of version

The querying of versions of bioinformatics tools/scripts and databases of a GitHub or non-GitHub project is the basic function of BioInstaller. For GitHub projects items, the GitHub APIs were used to access the projects version information, such as release, tags, and branches. All released versions will be used as the available versions and returned to BioInstaller (Fig. S1A). However, the situation becomes more complicated if the resources have not been published on GitHub. Here, we propose two types of methods of

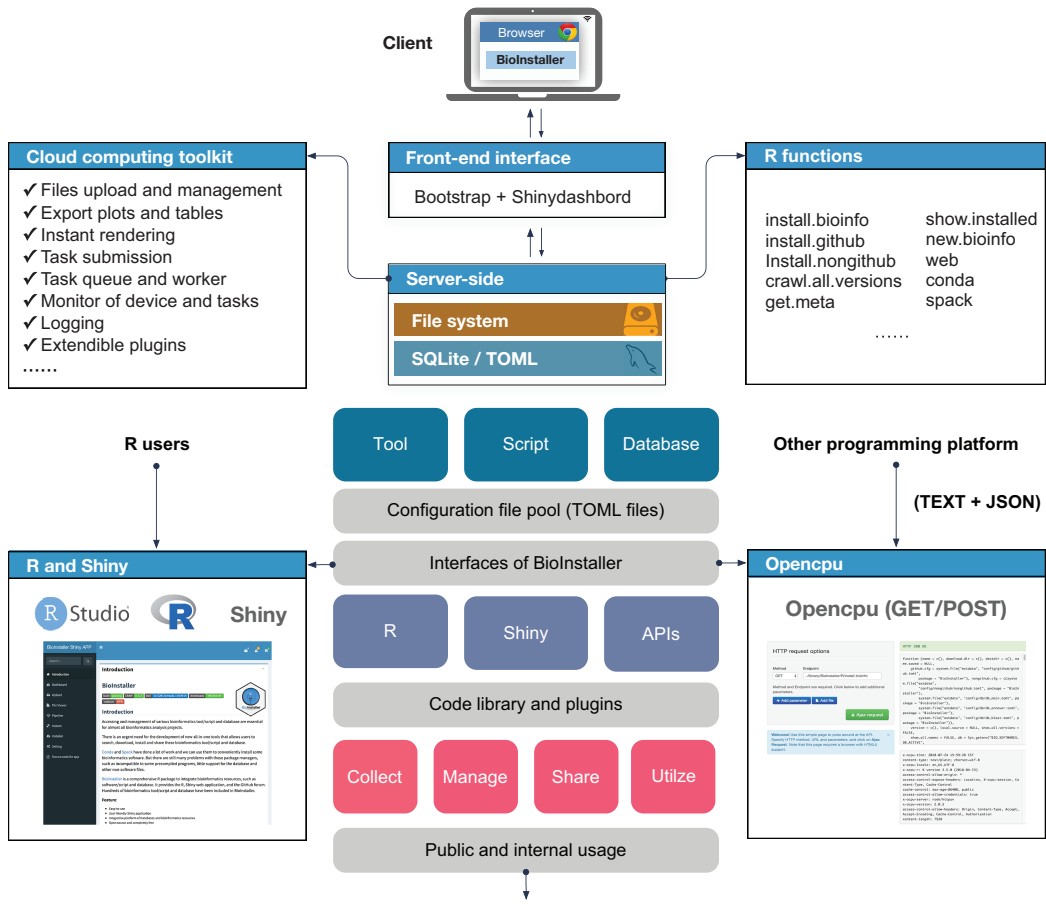

**Figure 1 Overview of structure and functions of BioInstaller.** Bioinformatics tools, scripts and databases are supported by BioInstaller. Bootstrap and Shinydashbord are used to construct the front-end interface. The R functions, Shiny and Opencpu services and the SQLite and TOML databases were applied in the back-end.

parsing item versions. Method I: If the released versions are fixed, users can write it in the "version_available" field in the configuration file. Method II: Utilizing the R packages rvest (https://CRAN.R-project.org/package=rvest) and RCurl (https://CRAN.R-project. org/package=RCurl), we established an R functions pool to dynamically query the version of items from the original release website (Dataset S1). The demo function to query the latest version of GMAP is shown in Fig. S1B. This is useful for automating a pipeline to build the precompiled binary version.

## Mirror resource for an invalid link

Network transferring is a common problem in bioinformatics data analysis. A mirror resource is one option to partially resolve these problems, including an invalid link and network blocking. BioInstaller allows users to set any numbers of mirror URLs for their tools/scripts and databases to avoid the possible problems caused by network transmission. As shown in Fig. S1C, the mirror URLs of Miniconda (https://conda.io/miniconda.html) are separately provided by the official and our hosts.

Notably, established mirror URLs of bioinformatics resources can be used in the spack (*Gamblin et al., 2015*) and other similar tools to build the cache files.

### TOML format configuration files

Massive bioinformatics tools/scripts and databases have been integrated into BioInstaller. TOML is a popular and human-readable configuration formats supporting comments. We uses standard TOML format configuration file to store required information of the included bioinformatics tools/scripts and databases. These configuration files can be reused in other bioinformatics software packages or data analysis pipelines via online accession or as a file copy. We have provided six directories to store different types of TOML files including "github," "nongithub," "database," "web," "docker," and "shiny." Due to the broad compatibility of BioInstaller, any resource published on docker, GitHub, Zenodo (https://zenodo.org/) or other platforms can be supported.

### Implementation of the Shiny application

To increase the convenience of BioInstaller for nonprogramming users, a user-friendly web application was developed based on Shiny (*Chang et al., 2015*). The user-interface (UI) of BioInstaller was constructed using the R package shinydashboard (https://cran.r-project.org/package=shinydashboard) and Shiny (*Chang et al., 2015*). Output tables were generated by the R package DT (https://CRAN.R-project.org/package=DT) and wrapped JavaScript library DataTables (https://datatables.net/). Charts were mainly generated by published R packages and in-house scripts or R packages that all support interactive update and export of PDF, SVG, and PNG format plots. The tab items of the BioInstaller Shiny application at the left side of the navigation bar can be used to switch among all available modules, including "Introduction," "Dashboard," "Upload," "File Viewer," "Pipeline," "Instant," "Installer," and "Setting." The detail usage guidelines are provided on our host (http://bioinfo.rjh.com.cn/labs/jhuang/tools/BioInstaller/), and R users can also use the browser vignettes functions in R to access these documents.

## RESULTS

### Overview and practices of BioInstaller's functionalities

A comprehensive R package was developed that could be used to quickly construct interactive and reproducible biological data analysis applications based on the R platform (Fig. 2). The functionalities (Table 1; Dataset S2) of BioInstaller were divided into six parts based on whether users use BioInstaller or not: (1) deployment of resources, (2) collection of resources, (3) sharing of resources, (4) construction of pipelines, (5) construction of Shiny applications, and (6) reproducible data analysis. An example of a real project (annovarR, https://github.com/JhuangLab/annovarR, under development) is shown in Fig. 2 to illustrate the full workflow for BioInstaller utilization, which was designed to integrate various genetic variant annotation and visualization tools, including public command line tools, R packages and custom annotation and visualization functions. Using the code library, predefined TOML files (database resources and plugins), and the docker file of BioInstaller, we could easily customize

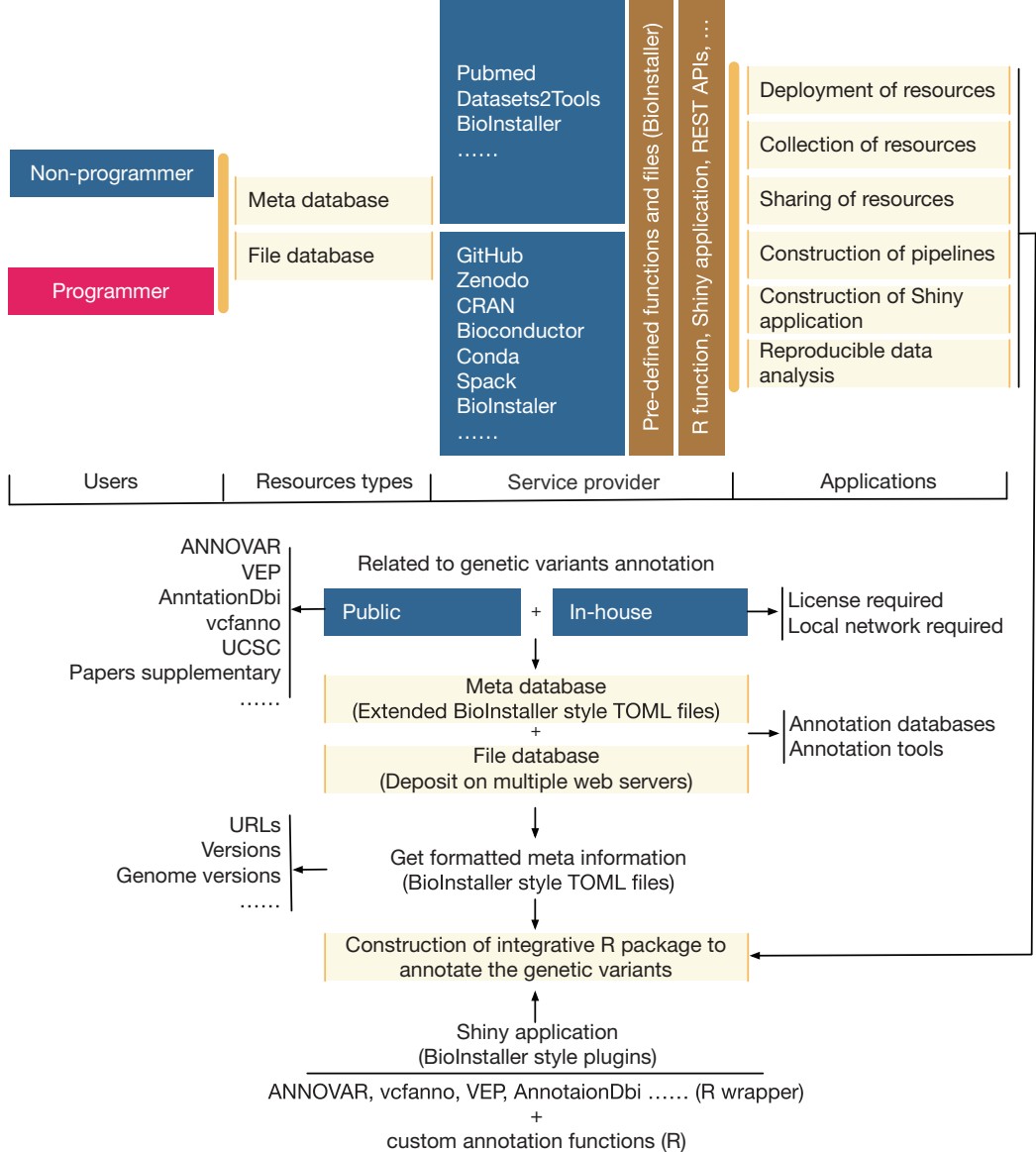

**Figure 2** **The relevance, applicability, and a real example of BioInstaller.**

the BioInstaller-established Shiny application to work on the genetic variants annotation tasks. If BioInstaller is not used, we need to develop the UI and server code of the Shiny application for a large number of universal functions, such as the file management system, background task submission and queue management, and tracking of the output log and files. The docker image of BioInstaller is also out-of-the-box and could be modified and applied to our own works. Based on the integrated installer (e.g., conda, spack, and BioInstaller) and simplified TOML files of BioInstaller, users can collect, share, and deploy genetic variant annotation databases and tools with one-stop service. As a real practice of BioInstaller, we collected and shared tools/scripts and databases in the configuration pool of BioInstaller, including genetic

**Table 1 List of the relevance and applicability of BioInstaller.**

| | With bioInstaller | Without bioInstaller |
|---|---|---|
| **Deployment of resources** | | |
| User-interfaces | R functions, Shiny UI, REST APIs (Conda, Spack, and other tools/scripts) | Command-line tools (Conda, Spack, and custom tools) |
| Retrieve installed packages | Integrated Shiny dashboard page including R packages, conda and Python packages, Spack packages, and BioInstaller resource | Multiple command line operations |
| **Collection of resources** | | |
| Local development | Yes | No |
| Need to register an account | Not need | Need |
| Type of backend databases | Default use TOML and SQLite (potable purpose) Plugins for other types | MySQL |
| **Resources hosts** | No limitation | Centralized |
| File sizes | No limitation | Limited |
| PubMed query | Integrated R codes with secret key (no limited access) Shiny UI with formatted table | Isolated R codes without secret key (limited access, $n <= 20$) Online version without formatted table |
| **Sharing of resources** | | |
| Medium | Simplified TOML format files | Form or configuration file required more skills |
| Download service | Local Shiny application | Centralized web service or command line tools |
| **Construction of pipelines** | | |
| Store of meta information (e.g., URL and version) | Pre-defined TOML file | *De novo* source code (e.g., ANNOVAR and fusioncatcher) |
| **Construction of Shiny application** | | |
| Pre-defined pages | Pre-defined Shiny UI and server (Dashboard, file management, task submission, logging, export, and update of plots exception handling, setting) | Isolated examples UI and server codes |
| Difficulty | Easy to construct the Shiny application (Plugins + optional R codes) | Relatively complicated (Require R codes for UI and server) |
| **Reproducible data analysis** | | |
| Logging | Support | Manual |
| Docker image | Pre-defined docker image with Shiny, Rstudio, and Opencpu services | Most not |

variant annotation databases and tools; the meta information is freely available and hosted on the public GitHub website (https://github.com/JhuangLab/BioInstaller/tree/master/inst/extdata/config). The raw files are stored on the original websites (e.g., https://github.com, https://sourceforge.net/, http://annovar.openbioinformatics.org/, etc.) and our host.

## Comparison of BioInstaller with existing tools for the collection and sharing of bioinformatics resources

To better understand the advance provided by BioInstaller in terms of the collection and sharing of bioinformatics resources, we further compared BioInstaller with several existing tools, including Omictools (*Henry et al., 2014*) and Datasets2Tools (*Torre et al., 2018*) (Tables 1 and 2), the two most comprehensive meta databases focused on bioinformatics tools. All provide a web forum to update the meta database of

**Table 2 Overview comparison of BioInstaller and existing tools on the collection and share of bioinformatics resources.**

|  | BioInstaller | Omictools | Datasets2Tools |
|---|---|---|---|
| **Infrastructure and utilities** |  |  |  |
| Programing language | R, JavaScript | HTML/CSS/ JavaScript | HTML/CSS/ JavaScript |
| Chrome extension | No | No | Yes |
| Web service | R Shiny | Web | Web |
| R functions | Yes | No | No |
| REST APIs | Yes | No | Yes |
| Backend database | TOML and SQLite | Not available | MySQL |
| Docker image | Yes | No | No |
| **Functionality** |  |  |  |
| Access and collect meta database | Yes | Yes | Yes |
| Access and collect file database | Yes | No | No |
| Integration of external resources | Yes | No | No |
| PubMed query | Yes | No | No |
| Dataset query | Yes | No | Yes |
| Number of supported resources | Integrated | High | Medium |
| Version query | Yes | No | No |
| Download service | Yes | No | No |
| Local branch and development | Yes | No | No |
| **Input and output** |  |  |  |
| Input | R functions, Web text, APIs | Web text only | Web text + APIs |
| Output | Text, table, plots, and Web page (PNG, SVG and PDF) | Web page | Text and web page |

bioinformatics resources. However, BioInstaller offers an off-line way to develop the users' own meta databases via an unlimited configuration file pool (TOML and SQLite format) that is easy to carry and share and is independent of programming knowledge. In addition, the developed R functions and Shiny application can be used to query and download the linked or isolated file databases, such as appendix data from papers, annotation databases for genetic variants, genome sequences, etc. In most cases, it is suitable to tightly combine the meta database with the file database. Therefore, we designed and shared an upload module in the Shiny application to set the meta information for all files, and users can add the description, genome version, custom file types, and other customizable fields. Both Omictools and Dataset2Tools only include the items in their databases and do not integrate external resources. BioInstaller not only can be used to collect users own resources, but also can be used to integrate external resources.

## Summary of supported bioinformatics tools/scripts and databases

For now, 157 tools/scripts and 110 databases are natively supported in BioInstaller (Fig. 1 and Table 3; Tables S1 and S2). First, we covered the most commonly used tools in

**Table 3 Summary of BioInstaller included tools/scripts and databases.**

| Category | Number |
|---|---|
| **1, Tools/scripts** | |
| Alignment and assembly | 27 |
| Quality control | 17 |
| HTS manipulation | 17 |
| Association analysis | 6 |
| Genetic variants annotation | 12 |
| Detection of SNVs, INDELs and SVs | 32 |
| Immunity-associated | 2 |
| Isoform analysis | 3 |
| Gene expression analysis | 9 |
| Network analysis | 3 |
| Visualization libraries | 11 |
| System dependence | 18 |
| **2, Databases** | |
| Variant-level | |
| Allele frequency | 17 |
| Variants Effect prediction | 29 |
| Disease-related | 6 |
| Gene-level | |
| Basic information | 8 |
| Gene function | 3 |
| Disease-related | 7 |
| Drug related | 4 |
| Noncoding RNA related | 15 |
| Reference genome | 9 |
| Protein related | 4 |
| Others | 8 |

each bioinformatics analysis process, including data quality control ($n = 17$), alignment and assembly ($n = 27$), variant detection ($n = 32$) or annotation ($n = 12$), high-throughput sequence manipulation ($n = 17$), and visualization libraries ($n = 11$) (Table 3; Table S1), etc. Second, BioInstaller also provides abundant databases for annotating data or satisfying software dependencies. With BioInstaller, users can easily download UCSC sequence and annotation data ($n = 4,995$) (Dataset S3), blast databases ($n = 29$) (Table S2), allele frequency databases ($n = 17$), variant effect prediction databases ($n = 29$), and disease-related ($n = 13$), drug-related ($n = 4$), noncoding region-related databases ($n = 15$) (Table 3; Table S2), among others. Notably, we collected and constructed 20 genetic variant annotation databases, which can be directly used in other variants annotation tools, including ANNOVAR (*Wang, Li & Hakonarson, 2010*), vcfanno (*Pedersen, Layer & Quinlan, 2016*), and annovarR (https://github.com/JhuangLab/annovarR).

BioInstaller has been released on CRAN for one and a half years and has accumulated a certain number of users, with a total of 19,912 downloads from CRAN (2018.8.3).

In the recent release (v0.3.5), we provided the Shiny application and significantly expanded the supported tools/scripts and databases. The number of supported tools/scripts and databases is still increasing and is being applied to other related projects, such as the integrated genetic variants annotation tool annovarR (https://github.com/JhuangLab/annovarR).

## Examples of BioInstaller R functions

We have demonstrated the basic structure, functions, and web service of BioInstaller. The full help document is available at http://bioinfo.rjh.com.cn/labs/jhuang/tools/BioInstaller/articles/. Because most of the Shiny application UIs are wrapped with R functions, we use several use examples to illustrate the R functions of BioInstaller.

**Example #1:** Install packed or unpacked bioinformatics tools. We use the Ion Torrent Variant Caller (*Zook et al., 2014*) and svaba (*Wala et al., 2018*) to show how to install or download the bioinformatics tools or scripts that are not supported by other package management tools.

```
> library(BioInstaller) # Library the R package
> set.biosoftwares.db("~/.BioInstaller/info.yaml") # Store the installation information
> install.bioinfo(show.all.names = TRUE) # Get all items name supported by BioInstaller
> install.bioinfo(name = "tvc," show.all.versions = TRUE) # Get all available versions of tvc
> install.bioinfo(name = "svaba," show.all.versions = TRUE) # Get all available versions of svaba
> install.bioinfo(name = "tvc," download.dir = "/path/tool/tvc") # One-click install the tvc
> install.bioinfo(name = "svaba," download.dir = "/path/tool/svaba") # One-click install the svaba
> show.installed() # Get all installed tools
> get.info("svaba") # Get the svaba installation information, such as update time and version
```

**Example #2:** Download genetic variants annotation databases. Genetic variants annotation is a common and high-demand task for most biomedicine research, especially for examining the correlations between phenotype and molecular features, such as germline and somatic mutations. The followed example describes how to download the genetic variants annotation databases dbSNP, CIViC, DisGeNET, and CancerHotspot (*Chang et al., 2016*; *Griffith et al., 2017*; *Piñero et al., 2017*).

```
> install.bioinfo("db_annovar_avsnp," extra.list = list(buildver = "hg19"), download.dir = "/path/db/") # install the latest dbSNP from ANNOVAR website
> crawl.all.versions("db_annovar_avsnp") # Download all dbSNP to current directory
> install.bioinfo("db_civic," download.dir = "/tmp/db") # Download the nightly version of CIViC database
> install.bioinfo("db_disgenet," download.dir = "/tmp/db") # Download the DisGeNET database
```

> install.bioinfo("db_cancer_hotspots," download.dir = "/tmp/db") # Download the DisGeNET databaseß

**Example #3:** Download an annotation database based on the supplementary files of published papers. The followed example is an epigenetic genes classification (e.g., reader, writer, eraser) database only available in the papers supplementary file (*Huether et al., 2014*).

> install.bioinfo("db_annovar_epi_genes," extra.list = list(buildver = "hg19"), download. dir = "/path/db/") # install the epigenetic genes database from our website

## User-interfaces and functions of the Shiny application

### Introduction module

Utilizing the Shiny function "includeMarkdown," we generated the "Introduction" module page from Markdown, a lightweight markup language, format document (Fig. S2A).

### Dashboard module

The "Dashboard" module includes the system monitors, such as hardware (Disk and memory), queue tasks, task log, installed R packages, Python packages, conda environments, and the other information of the operating environments (Figs. S2B, S2C and S3). The monitored data stream is automatically updated once every 10 s (Fig. S2B). A demo table output in the dashboard lists all files in the environment variable "PATH," where users can use the selector at the lower left quarter to customize the row numbers (5, 10, 25, 50, and all) (Fig. S2C). All output tables in BioInstaller can be easily exported to CSV, XLS, PDF files or directly copied to the clipboard. Monitor plugins related to the information of the R system (Figs. S3A, S3B and S3C), conda (Fig. S3D), BioInstaller (Fig. S3E), and spack (Fig. S3F) are integrated in this work, which can reduce user input of extra command line commands and facilitate sharing with others.

### Upload module

The "Upload" module is used to upload files to the BioInstaller Shiny web platform. Optional fields, such as file type, genome version, and description, can be stored in the SQLite format database with the uploaded files path and the files md5 value (Fig. S2D). When uploading a file, users need to click the "Save" button to confirm the upload behavior and update the database (Fig. S2E). Before the confirmation click, users can preview the file and make a final decision (Fig. S2F). Files with sizes ranging from 0.25 to 8 GB were tested on the Shiny application (Table S3). For files larger than 10 GB, we recommend using the rsync or FTP service to transfer files and then adding the corresponding description and records in TOML or SQLite databases.

### File viewer module

The "File viewer" module is used to manage all uploaded files in the BioInstaller Shiny application that supports view, delete and download, and all files can be used in the other plugins of the BioInstaller Shiny application, mainly in the "Pipeline" and "Instant" modules (Figs. 3A and 3B).

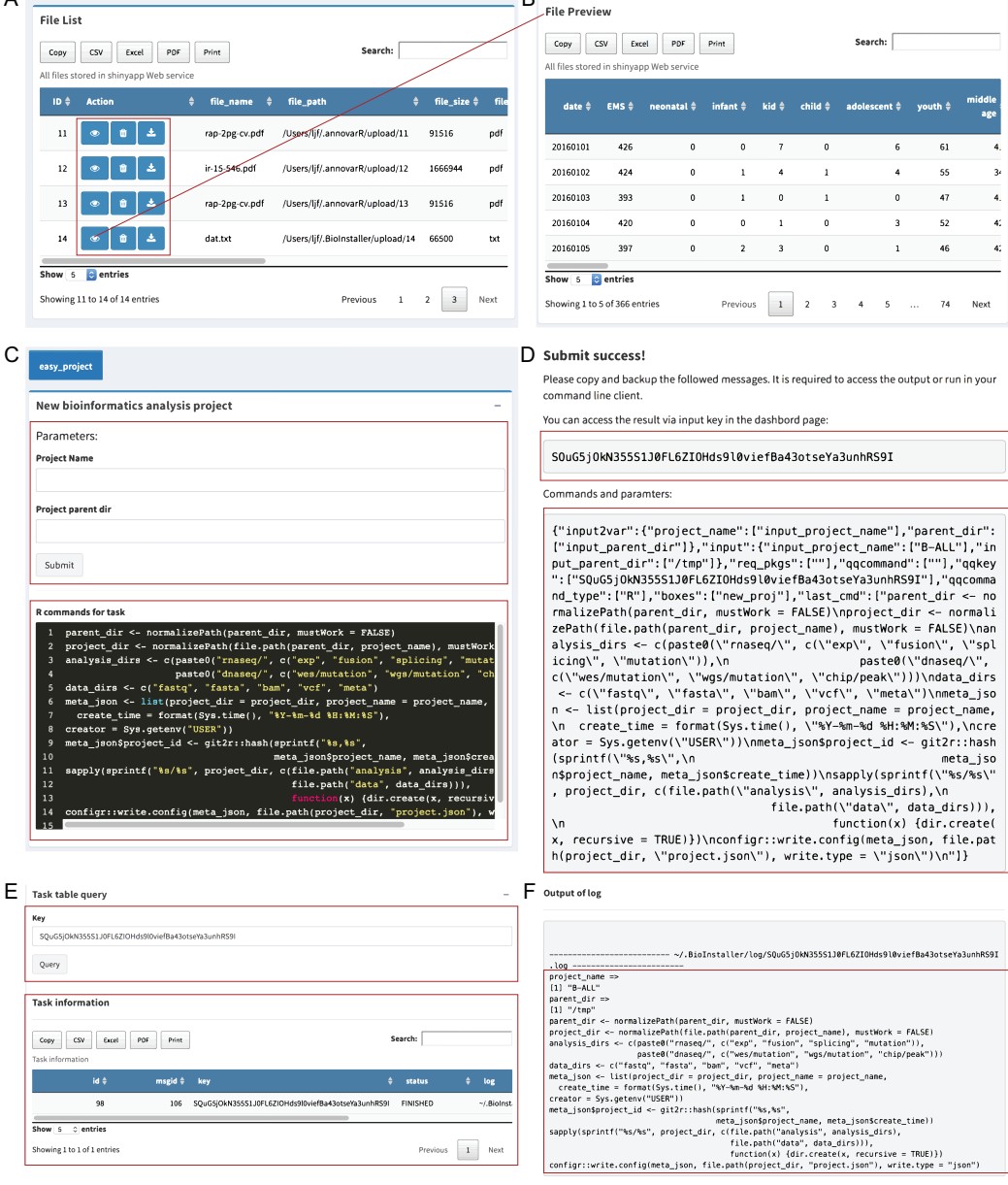

**Figure 3 Shiny application modules of file viewer and pipeline.** (A) Uploaded files are showed on the table where view, deletion, and download function are provided. (B) The interface of the preview result. (C) Easy project was used as the demo in pipeline module, which could be used to create a series of directories via submitting a queue task with two parameters: project name and parent directory. (D) The dialog box displays a prompt message with a queue character key. (E) Task queue and queue information can be requested by the character key in the dashboard module. (F) Function to get the output log of the submitted task.

## Pipeline module

The "Pipeline" module is used to integrate complicated bioinformatics analysis workflows or other small scripts. An in-house interpreter R function was used to parse the plugin configuration files to generate the Shiny UI and server functions. A small script creating a data analysis directory structure was used as the demo for "Pipeline"

(Dataset S4). Users can input the project name and the parent directory to create a series of predefined directories. The R commands used in this task are editable at the bottom of the box (Fig. 3C). After users click the "Submit" button, BioInstaller will generate a random character as the submitted task key. Users can use this key to retrieve the output information, such as files and logs, in the "Dashboard" module (Figs. 3C–3F). All submitted tasks enter the task queue supported by the SQLite database using the R package litseq (https://CRAN.R-project.org/package=liteq). Tasks in the queue are automatically checked by the activated workers (Figs. 3E and 3F).

### Instant module

The "Instant" module is used to run the real-time plots and data analysis, and similar to the "Pipeline" module, the UI and server were automatically generated via plugin configuration files (Dataset S5). We used the meta database query of BioInstaller, Datasets2tools (*Torre et al., 2018*), PubMed, and plots of Maftools (*Mayakonda & Koeffler, 2016*), a cancer somatic mutations visualization tool, as the demo to demonstrate the function. Users can select the input files defined in the plugins configuration file (TOML) or user-uploaded files. The commands are stored in the bottom of the boxes and can be modified by the user. After clicking the "Run" button, all output box codes, such as output plots and tables, run on the server side and are returned in real time to the Shiny UI (Fig. 4). We developed several plugins to query and access several meta databases related to bioinformatics, such as the BioInstaller meta databases (Figs. 4A and 4B), Datasets2tools (*Torre et al., 2018*) (Figs. 4C and 4D), and PubMed (Figs. 4E and 4F). The powerful visualization functions of R packages are also supported in the "Instant" module. As shown in Figs. 4G and 4H, users can obtain the demo output (PDF and PNG format) of Maftools. After running all box codes, a single box can be separately updated and exported by users.

### Installer module

The "Installer" module is the main Shiny interface of BioInstaller for downloading and installing bioinformatics tools/scripts and databases. We provide the Shiny interfaces of BioInstaller, conda and spack (Fig. S4). The "Installer" module is similar to the "Pipeline" module, which is also needed to submit a task to the queue. The status and log information can be retrieved in the "Dashboard" module. Three basic use cases of the BioInstaller Shiny application are available: (1) download db_annovar_refgene database (Fig. S4A–S4B); (2) create conda environment (Fig. S4C–S4D); (3) install "zlib" using spack (Fig. S4E–S4F).

### Setting module

The setting module is the interface for setting the value of the variable used in the BioInstaller plugins or R files. Both a Shiny UI and a YAML editor are offered for users (Figs. S5A and S5B). Any updates of the YAML editor (Fig. S5B) can change and refresh the Shiny UI options (Fig. S5A). It is helpful for users to manage various material related to BioInstaller and its plugins.

In most cases, through the one-click interface of BioInstaller, users can easily download and install the desired bioinformatics resources without any command line skills. Functions for automatic compiling from the source file with the dependent software

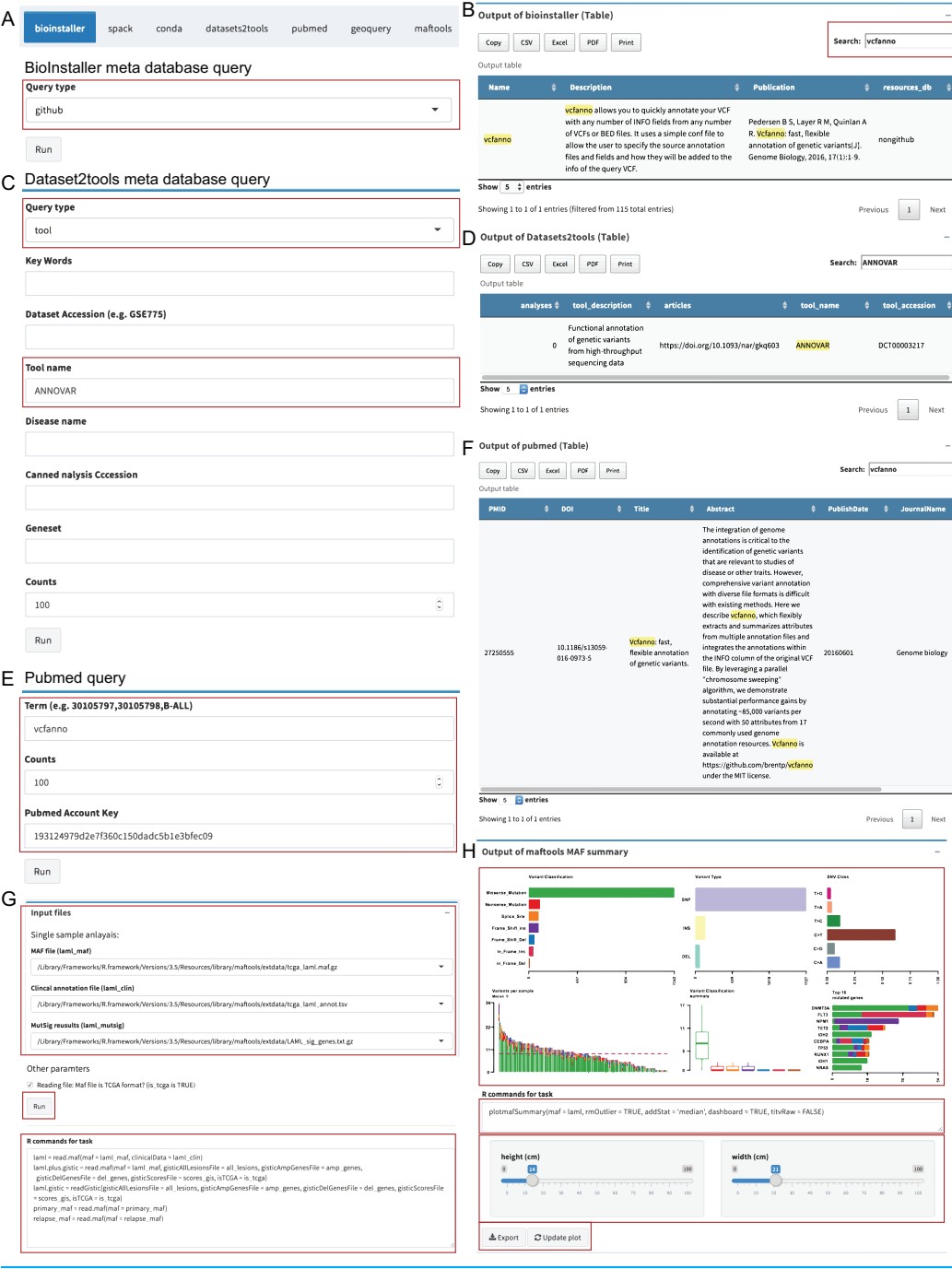

**Figure 4  Shiny application "Instant" module.** (A, B) The input box and output of BioInstaller Shiny plugin for querying the TOML format meta databases. (C, D) The input box and output of Dataset2tools Shiny plugin. (E, F) The input box and output of PubMed query Shiny plugin. (G) The input box of Maftools Shiny plugin. (H) One of Maftools demo output boxes which contains the plot, R command, export and update button.                                                           

or database are also supported in BioInstaller. However, for complicated software with high system dependence, we recommend using the interfaces of conda (https://conda.io/docs/) and spack (*Gamblin et al., 2015*).

## Portable message queue for background tasks based on SQLite

Tasks with long-time costs are challenging in Shiny, which always blocks the other interactive operations simultaneously when the previous task has not been finished. Here, we utilized the R package litseq (https://CRAN.R-project.org/package=liteq) to submit and manage the background queue tasks. litseq is portable and lightweight. litseq does not require extra software or service from other programming platforms and can work on any clusters server running computing-intensive tasks. The developed queue worker in BioInstaller can be used for all other background tasks submitted by litseq. All litseq-submitted tasks of BioInstaller are assigned a unique identification id. All executed commands, output logs, and others are saved in the permanent files.

## Opencpu backend service improves reproducibility

Opencpu (*Ooms, 2017*) is an R package for reproducible research that can expose a web REST API interface with R, Latex, and Pandoc. The R functions of BioInstaller are invoked by the activated Opencpu R process or daemon service. For other programming platform users, this is one possible method for utilizing the R functions of BioInstaller (Fig. 1). The output of JSON and text formats are returned when using the browser access (Fig. 5A) or simulated requests. Three of the most basic APIs usages of BioInstaller were used to demonstrate how it works: (1) obtaining all supported tools/scripts and databases; (2) acquiring available versions of the appointed item; (3) installing a tool in a directory (Fig. 5B). Notably, a random string, such as "x0a469794fa," will be generated as the key of Opencpu to obtain the output of one R session. Both JSON and text format output can be returned by Opencpu backend APIs (Fig. 5C).

## Docker container of BioInstaller

A prebuilt docker image is available on the DockerHub (https://hub.docker.com/r/bioinstaller/bioinstaller), and the latest code change of the BioInstaller repository can automatically trigger an update of the docker image. In the docker image, we integrated and configured three types of web services, including Opencpu, Shiny (*Chang et al., 2015*; *Ooms, 2017*), and the RStudio server (https://www.rstudio.com/products/rstudio-server/). The followed commands can be used to deploy and start the service of BioInstaller service.

```
$ docker pull bioinstaller/bioinstaller
$ docker run -it -p 80:80 -p 8004:8004 bioinstaller/bioinstaller
```

Users can deploy a new instance host of BioInstaller and all other web services in a few minutes, and other tools/scripts and databases are also allowed to be embedded in this docker image using the Dockerfile (https://github.com/JhuangLab/BioInstaller/blob/master/Dockerfile).

## Use the GitHub forum to share, rate, and discuss the bioinformatics resources

The full-text search is natively supported by the GitHub website with highlight and age forwarding functions (Figs. S6A and S6B). To simplify the submitting of

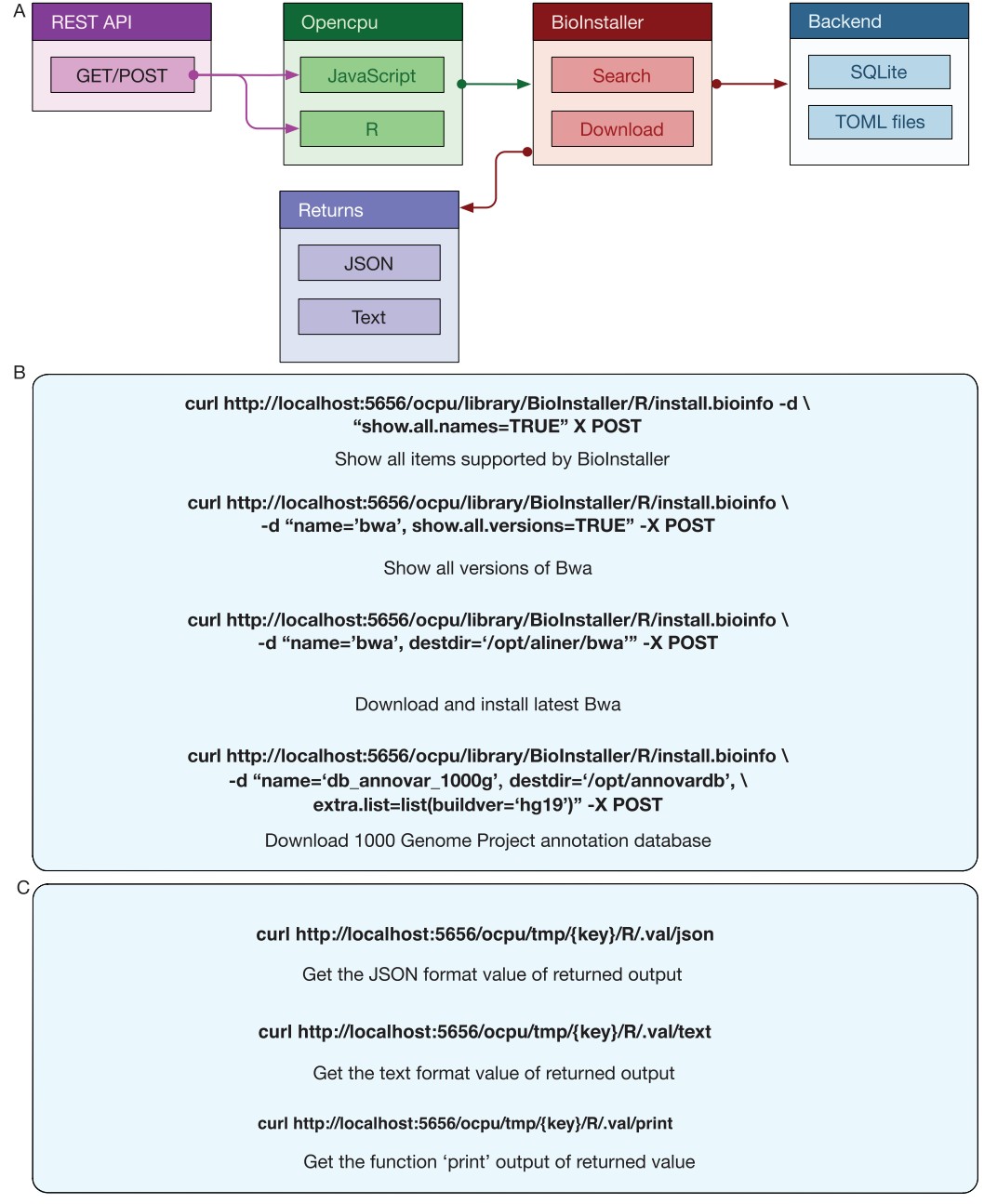

Figure 5  **REST APIs of BioInstaller.** (A) Workflow of REST APIs of BioInstaller that JSON and TEXT returns through the GET/POST query. (B) Using curl to invoke background R functions of BioInstaller. (C) The key character with GET method is provided to get the background R session output.

new items to BioInstaller, the GitHub repository issues page (https://github.com/JhuangLab/BioInstaller/issues) is recommended for other users to share, rate, and discuss bioinformatics tools/scripts and databases with a designated label (Fig. S6C). The "watching" function of GitHub can allow users to receive notifications of all conversations on the BioInstaller. Another advantage of establishing a free sharing

community based on the GitHub is that all history changes on the code and forum posts can be recorded and retrieved. A rating function for bioinformatics tools/scripts or databases is also feasible by calculating the points corresponding to thumbs up or down.

## DISCUSSION

Bioinformatics tools/scripts and databases are widely used in various data analysis projects. The construction of interactive and reproducible biological data analysis applications is critical for most bioinformatics data analyses (*Henry et al., 2014*; *McQuilton et al., 2016*; *Ohno-Machado et al., 2017*). The integrative utilization of these resources is becoming increasingly important for improving integrated biosciences data analysis. R language, as the most popular programming language for statistics, biological data analysis, and big data, has provided massive useful R packages for various data analysis efforts, especially the NGS field. However, there has been no comprehensive and free R application that can support file upload and management, perform long-time computation with a tasks submission system, track and record the output of files and log, develop extendible plugins, add or remove functions of the application in real time, and respond for REST APIs. Another common problem for users of R and other programming platforms for biological data analysis is that massive bioinformatics resources are isolated and scattered, which significantly increases the difficulty of deploying, collecting and sharing these resources. Well-known software distribution tools that do not need root privileges, such as conda (https://conda.io/docs) and spack (*Gamblin et al., 2015*), were designed for comprehensive fields and usually lack support for life science resources. Bioconda is a fine example of the centralized installation of bioinformatics software (approximately 1,900 items) that has significantly improved the reproducibility of bioinformatics data analysis (*Gruning et al., 2018*). However, this is not sufficient compared with the rapid increase in software and databases in the life sciences field.

As described in this study, we present a comprehensive, free and open-source platform, BioInstaller, to construct the interactive and reproducible biological data analysis applications. BioInstaller contains the R functions, the Shiny application, REST APIs and the docker image. This platform and the practices described in this work are sufficient for most R users to conveniently and quickly develop an interactive and reproducible biological data analysis application with diverse predefined functions (e.g., file management, task submission, plugin management system, logging, etc.), plugins, and files offered by BioInstaller. Moreover, based on the TOML format files, we have also integrated hundreds of bioinformatics resources required for the wide field of bioinformatics, such as sequence alignment, variant calling and annotation, and so on. We hope this newly presented open-source platform for R users can reduce the difficulty of constructing the interactive and reproducible biological data analysis applications and further improve the interactivity and reproducibility of bioinformatics data analysis.

## CONCLUSION

As described in this work, we established a new platform to construct interactive and reproducible biological data analysis applications based on R language. This platform contains diverse UIs, including the R functions and R Shiny application, REST APIs, and support for collecting, managing, sharing, and utilizing massive bioinformatics tools/scripts and databases.

## ACKNOWLEDGEMENTS

We would like to thank the many BioInstaller users who provided feedback and suggestions. We thank the Centre for HPC at Shanghai Jiao Tong University for supporting computing platforms. We thank the Samuel Waxman Cancer Research Foundation for supporting followed research.

### Funding

This work was supported by the National Natural Science Foundation of China (No. 81570122, 81770205), the National Key Research and Development Program (No. 2016YFC0902800), and the Shanghai Municipal Education Commission-Gaofeng Clinical Medicine Grant Support (20161303). The funders had no role in study design, data collection and analysis, decision to publish, or preparation of the manuscript.

### Grant Disclosures

The following grant information was disclosed by the authors:
National Natural Science Foundation of China: 81570122, 81770205.
National Key Research and Development Program: 2016YFC0902800.
Shanghai Municipal Education Commission-Gaofeng Clinical Medicine Grant Support: 20161303.

### Competing Interests

The authors declare that they have no competing interests.

### Author Contributions

- Jianfeng Li conceived and designed the experiments, performed the experiments, analyzed the data, contributed reagents/materials/analysis tools, prepared figures and/or tables, authored or reviewed drafts of the paper, approved the final draft.
- Bowen Cui conceived and designed the experiments, performed the experiments, analyzed the data, contributed reagents/materials/analysis tools, prepared figures and/or tables, authored or reviewed drafts of the paper, approved the final draft.
- Yuting Dai contributed reagents/materials/analysis tools, authored or reviewed drafts of the paper, approved the final draft.
- Ling Bai contributed reagents/materials/analysis tools, authored or reviewed drafts of the paper, approved the final draft.

- Jinyan Huang conceived and designed the experiments, contributed reagents/materials/ analysis tools, authored or reviewed drafts of the paper, approved the final draft.

## Data Availability

Project name: BioInstaller

Project home page - Package: http://bioinfo.rjh.com.cn/labs/jhuang/tools/bioinstaller

Project home page - Package (CRAN): https://CRAN.R-project.org/package=BioInstaller

Project source code - Package: https://github.com/JhuangLab/BioInstaller.

Project docker image: https://hub.docker.com/r/bioinstaller.

Operating system(s): Platform independent

Tested Operating system(s): MAC OS 10.13.6, Centos 7.1, Ubuntu 16.04, and Windows 10 (limited functions)

Tested browsers: Google Chrome (v68), Safari (v11.1.2), and Firefox (v61.0.2)

Programming languages: R, HTML, CSS and JavaScript

Requirements: R (> 3.3.0), Shiny (R package), Opencpu (R package), and a web browser

Optional functions: Maftools (R package), conda (Python package), spack (Python package), and docker

License: MIT.

## Supplemental Information

Supplemental information for this article can be found online at http://dx.doi.org/10.7717/peerj.5853#supplemental-information.

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
