# Peer review of "BioInstaller: a comprehensive R package to construct interactive and reproducible biological data analysis applications based on the R platform"

_PeerJ, doi:10.7717/peerj.5853_

## Round 0.1 · original submission · Major Revisions

Dear Dr. Li and colleagues:

Thank you for submitting your manuscript to PeerJ. I have now received two independent reviews of your work, and as you will see, the reviewers raised some concerns about the research. Despite this, these reviewers are optimistic about your work and the potential impact it will have on the computational biology community. Thus, I encourage you to revise your manuscript accordingly, taking into account all of the concerns raised by both reviewers.

In your revision, please try to discuss the relevance and applicability of BioInstaller. Obviously, this is important to attract the attention of a wide range of biologists that would benefit from the programs. One important topic that is missing is how BioInstaller overcomes limitations encountered with other approaches. Please address this, and provide a work flow with and without BioInstaller that shows how your programs optimize the work flow. Also, be sure to provide some working examples and reports on the download time for datasets of varying sizes.

I look forward to seeing your revision, and thanks again for submitting your work to PeerJ.

Good luck with your revision,

-joe

Reviewer 1 ·

Basic reporting

This manuscript needs a lot of improvement in English writing. Grammar errors examples are:
1. line 142, 'what more' should be 'what's more'.
2. line 144, 'and so on' should be 'et. al'.
3. line 145, 'combined' should be 'combine'.
4. line 167, 'download' should be 'downloaded'

Improvement is also needed for wording, for example, in line 123, it's not appropriate to use 'non-professionals' to define users, maybe 'non-programming' is better. For 'Use case # xx' , it's better to change as 'Example # xx'.

Experimental design

The methods are well-explained and illustrated. However, the research question is not clearly specified in materials and methods. The authors should explain why it is necessary to develop such a tool to integrate download, install and share of bioinformatics tools/databases. What kind of issues are not solved by existing tools?

In line 74-75, 'integration of bioinformatics software/script and database' is confusing. It sounds like a pipeline providing bioinformatics analysis by combining software and database. Authors better explain the goal of this tool before introducing the design.

Validity of the findings

The authors didn't provide examples for uploading/sharing tools/data, and didn't report the time/speed of downloading/uploading a large dataset.

·

Basic reporting

The manuscript is well written with enough details and reference.

Experimental design

The technical design of the R package is comprehensive and novel.

Validity of the findings

The four cases well demonstrate the usage of the package.

Additional comments

The authors make great efforts to create a comprehensive R package, BioInstaller, with both Shiny application, and the HTTP representational state transfer (REST) APIs. The manuscript and the package are well written.

I only have one comment. Though enough technical details have been presented in the manuscript, Could the authors comment how the BioInstaller could be used for biological scientific finding?

---

## Round 0.2 · accepted · Accept

Dear Dr. Li and colleagues:

Thanks for re-submitting your manuscript to PeerJ, and for addressing the concerns raised by the reviewers. I now believe that your manuscript is suitable for publication. Congratulations! I look forward to seeing this work in print, and I anticipate it being an important bioinformatics resource for a broad community. Thanks again for choosing PeerJ to publish such important work.

Best,

-joe

# Reviewer 1 ·

Basic reporting

No comment

Experimental design

No comment

Validity of the findings

No comment

Additional comments

No comment

·

Basic reporting

The English writing is good.

Experimental design

The experimental design is comprehensive and reasonable.

Validity of the findings

The findings are validated.

Additional comments

The author addressed my comments.